# Using Contactless Facial Image Recognition Technology to Detect Blood Oxygen Saturation

**DOI:** 10.3390/bioengineering10050524

**Published:** 2023-04-26

**Authors:** Jui-Chuan Cheng, Tzung-Shiarn Pan, Wei-Cheng Hsiao, Wei-Hong Lin, Yan-Liang Liu, Te-Jen Su, Shih-Ming Wang

**Affiliations:** 1Department of Electronic Engineering, National Kaohsiung University of Science and Technology, Kaohsiung 80782, Taiwan; eagle@nkust.edu.tw (J.-C.C.); i109152108@nkust.edu.tw (T.-S.P.); i110152102@nkust.edu.tw (W.-H.L.); f109152127@nkust.edu.tw (Y.-L.L.); sutj@nkust.edu.tw (T.-J.S.); 2Division of Gastroenterology (General Medicine), Department of Internal Medicine, Yuan’s General Hospital, No. 162, Cheng Kung 1st Rd., Lingya District, Kaohsiung 80249, Taiwan; swc20342034@gmail.com; 3Department of Telecommunication Engineering, National Kaohsiung University of Science and Technology, Kaohsiung 80782, Taiwan; 4Department of Computer Science and Information Engineering, Cheng Shiu University, Kaohsiung 833, Taiwan

**Keywords:** facial recognition, non-contact detection, blood oxygen saturation

## Abstract

Since the outbreak of COVID-19, as of January 2023, there have been over 670 million cases and more than 6.8 million deaths worldwide. Infections can cause inflammation in the lungs and decrease blood oxygen levels, which can lead to breathing difficulties and endanger life. As the situation continues to escalate, non-contact machines are used to assist patients at home to monitor their blood oxygen levels without encountering others. This paper uses a general network camera to capture the forehead area of a person’s face, using the RPPG (remote photoplethysmography) principle. Then, image signal processing of red and blue light waves is carried out. By utilizing the principle of light reflection, the standard deviation and mean are calculated, and the blood oxygen saturation is computed. Finally, the effect of illuminance on the experimental values is discussed. The experimental results of this paper were compared with a blood oxygen meter certified by the Ministry of Health and Welfare in Taiwan, and the experimental results had only a maximum error of 2%, which is better than the 3% to 5% error rates in other studies The measurement time was only 30 s, which is better than the one minute reported using similar equipment in other studies. Therefore, this paper not only saves equipment expenses but also provides convenience and safety for those who need to monitor their blood oxygen levels at home. Future applications can combine the SpO2 detection software with camera-equipped devices such as smartphones and laptops. The public can detect SpO2 on their own mobile devices, providing a convenient and effective tool for personal health management.

## 1. Introduction

The currently available blood oxygen monitors are all contact based, with high equipment prices. Further, they require power, are inconvenient to carry, and often cause discomfort when in contact with the skin. This paper uses facial detection to determine SpO2, which has the advantages of being non-contact, fast, and highly accurate, allowing COVID-19 patients to monitor their own physical condition at any time and improve their health management.

This paper utilizes a simple type of network camera to capture a person’s area of interest and conduct photoplethysmography (PPG) calculations on the forehead features using the camera to obtain values for the R and B channels of the photoplethysmogram. After obtaining the values, the paper performs blood oxygen calculations and measures the light intensity, taking into account the effect of the strength of light and the brightness of the background on the experimental results.

### 1.1. Literature Review

References [1,2,3,4] use Raspberry Pi kits to capture and process facial images at a relatively low cost compared to other software and hardware devices in terms of RAM, processor capacity, and storage space. The Eulerian Video Magnification (EVM) algorithm is used to observe changes in skin color, which can detect time changes in images that are imperceptible to the naked eye. This method magnifies the captured image and performs spatial filtering on each frame of the image, and then amplifies the frequency band through temporal bandpass filtering before synthesizing the magnified image to observe the blood flow on the face. The blood oxygen formula is determined by linear regression using the research results of volunteers. The paper includes two calibration images and two detection images with a 5 min interval and an error rate of approximately 2% to 3%, which is more time-consuming than other methods but has similar accuracy.

References [5,6,7,8] use two types of charge-coupled devices (CCD) with narrowband filters of 520 nm (green light wavelength) and 660 nm (red light wavelength) to effectively resist interference from other wavelength bands of ambient light. They compare the wavelengths of 520 nm and 660 nm in the formula and record the test subject’s face at a speed of 25 frames per second using the recorded image as input. They also use a blood volume pulse (BVP) sensor on the finger to measure the relevant values. Although CCD can effectively resist other wavelength bands and can add passive components to address insufficient light sources, its price is much higher than that of general network cameras, not to mention the additional cost of adding other narrowband filters. This is a significant financial burden for hospitals or other long-term care facilities.

References [9,10,11,12] convert the RGB color space to the YCgCr color space. Y represents the luminance component, while Cg and Cr represent the chroma components for green and red, respectively. In order to capture relevant signals, a bandpass filter is applied to filter out relevant signals, followed by extracting the peaks and troughs of the Cr and Cg waves. By comparing the waveforms, the RCgCr is obtained, as shown in Figure 1. Subsequently, the obtained RCgCr value is calibrated using a linear regression between the RCgCr value and the reference SPO2 level. This document also compares YCgCr with YCbCr in the optical band, and the computational results show that YCgCr has a lower root-mean-square error by approximately 1 compared to YCbCr.

The experimental distance for this paper is 60 cm, and any distance exceeding this range may affect the experimental results. Due to the influence of light on the experimental results, the paper was conducted only under the illumination of 300–650 lux. During the experiment, it is important to avoid body movement to prevent excessive errors in the experimental data.

### 1.2. Standards for Blood Oxygen Saturation

Blood oxygen saturation is a fundamental element in the management and understanding of patient care. Oxygen is tightly regulated in the body, as hypoxemia can cause many acute adverse effects on individual organ systems. These include the brain, heart, and kidneys. Measurement of blood oxygen saturation compares the current amount of hemoglobin that is bound to oxygen versus that which is unbound. At the molecular level, hemoglobin is composed of four globular protein subunits, each of which is associated with a heme group. Each hemoglobin molecule subsequently has four heme binding sites that can bind oxygen molecules at any time. Therefore, in the process of transporting oxygen in the blood, hemoglobin is able to carry up to four oxygen molecules. Due to the critical nature of oxygen consumption in body tissues, it is essential to monitor the current blood oxygen saturation level [13].

Blood oxygen machines are used for mild COVID-19 patients who are quarantined at home or in isolation centers, as well as for asymptomatic individuals who are being monitored due to their exposure history. Some patients experience a rapid decrease in blood oxygen saturation, and signs of hypoxia may not be obvious, making a blood oxygen machine a useful tool in these situations. The New York State Department of Health recommends measuring blood oxygen saturation three times a day [14], as shown in Table 1. If the SpO2 measurement is equal to or greater than 95% each time, the blood oxygen concentration is considered normal. If the measurement falls between 91% and 94%, the individual should monitor their condition. If the SpO2 measurement is less than 90%, medical attention should be sought.

If the blood oxygen saturation is 94% or if it drops rapidly by more than 3% from the normal value, it means that there is something abnormal in some parts of the body. Usually, symptoms that can cause low blood oxygen include obstruction of the respiratory tract by a foreign object, cardiovascular disease, respiratory disease, surgical trauma, etc. When low blood oxygen occurs, the body may feel fatigued, have difficulty breathing, dizziness, weakness, rapid heartbeat, and inability to concentrate. In severe cases, it may lead to death. If the blood oxygen saturation is insufficient for a long time, it can cause rapid degradation of organ function, such as the brain, heart, and kidneys. Therefore, for bedridden patients or those with chronic diseases, checking blood oxygen saturation is an important indicator to observe the vital signs.

## 2. Research Method

### 2.1. Face Detection Technology 

With the advancement of technology, artificial intelligence (AI) has become increasingly popular in daily life. AI technology allows machines to perform human actions, such as facial recognition and face detection. These technologies not only improve people’s quality of life, but also increase their work efficiency. For example, inexpensive cameras can be used to create a monitoring system or even as a door lock. Most commercially available cameras come with object detection, including mask detection, face detection, and other related functions. These features can capture image characteristics to let us know when a person appears without having to monitor the image constantly. Face detection is a type of object detection that focuses only on detecting faces, usually using rectangular frames to indicate the position of the face or even more detailed features such as the eyes, forehead, and mouth. Commonly used face feature detection technologies include region of interest (ROI), Dlib facial landmarks, Haar features, and local binary patterns (LBP).

### 2.2. Region of Interest (ROI) 

A region of interest (ROI) refers to the area of an image that is extracted for processing using irregular polygons, ellipses, circles, rectangles, etc. Various operators and functions can be used to determine the ROI and process the subsequent image. ROIs are widely used in image segmentation, facial recognition, heat maps, and other fields, as shown in Figure 2. If we want to detect eyes in an image, we must first perform face detection on the entire image. When extracting the face image, only the facial area is selected, and the position of the eyes is searched for, rather than capturing the entire image. This approach improves accuracy and efficiency, and in general, ROI regions can be directly detected using pixel matrices.

The reverse propagation of regular ROIs [16] is shown in Equation (1):(1)$$\frac{\partial L}{\partial x_{i}}=\underset{r}{\sum }\underset{j}{\sum }\left[i=i^{*}\left(r,j\right)\right]\frac{\partial L}{\partial y_{rj}}$$*L* is the output layer, $x_{i}$ represents the pixel point of the feature map before pooling, $y_{rj}$ represents the *j*-th point of the r-th region after pooling, and $i^{*}\left(r,j\right)$ represents the coordinate of the highest pixel value selected during max pooling. According to Equation (1), only when a pixel value after pooling corresponds to the value of the current $x_{i}$ will the gradient be returned to$\;x_{i}$.

### 2.3. Dlib Facial Landmark Points

The histograms of oriented gradients (HOG) + linear support vector machine (SVM) algorithm in Dlib [17,18,19] can quickly recognize frontal faces. HOG feature extraction technique accumulates gradient orientations in each block to form the feature of that block. SVM is used to find a hyperplane that separates different classes of data.

The process of HOG feature extraction is shown in Figure 3 [17].

Image preprocessing involves first extracting the HOG feature extraction screen from the original image, with a size of 64 × 128 for best results, as suggested in reference [21,22]. To avoid color distortion when the image is reduced in size, gamma correction is applied. Calculating the gradients on the image can highlight the edges and contours, while smoothing areas without contour information can be eliminated through gradient calculation. Since the gradient values in smooth areas are generally small, they do not have much impact on the subsequent calculations.

Figure 4 shows the X-axis gradient on the left and the Y-axis gradient on the right, which are used to calculate the image gradient. The gradient is obtained by combining the magnitude and direction of the image. Firstly, the X-axis gradient Gx and Y-axis gradient Gy are calculated for each pixel using Formula (2) and Formula (3), respectively.
Gx(r,c) = I(r,c + 1) − I(r,c − 1)(2)
Gy(r,c) = I(r − 1,c) − I(r + 1,c)(3)
r and c, respectively, refer to row and column, and I is the image.

After calculating Gx and Gy the magnitude and orientation of the gradient for each pixel are computed using Equations (4) and (5).
(4)$$\mathrm{Magnitude}(\mu )=\sqrt{{\mathrm{Gx}}^{2}+{\mathrm{Gy}}^{2}}$$
(5)$$\mathrm{Angel}(\theta )=|\tan^{-1}\left(\frac{\mathrm{Gy}}{\mathrm{Gx}})\right|$$

In RGB images, the gradient with the highest intensity is selected as the main image, and its direction is calculated based on the gradient intensity of the maximum channel. After calculating the gradient, the expected features are extracted using blocks for analysis. This approach not only avoids the influence of single pixel values but also reduces noise interference.

The gradient angles are composed of nine feature vectors, including 0, 20, 40, 60, 80, 100, 120, 140, and 160 degrees. Since the intensity of 181 to 360 degrees is the same as that of 0 to 180 degrees, only the direction differs. Each pixel’s gradient intensity is placed in a specific gradient direction, as shown in Figure 5 [23,24,25]. After calculating each unit, the gradient intensities of each direction are summed up.

Due to the influence of image brightness, gradient calculation is susceptible to changes in light and shadow, with greater intensity for brighter images and lesser intensity for darker images. Therefore, normalization is necessary to reduce the effect of image brightness. The normalization is performed by dividing the image into blocks of 4 cells, and each block is normalized instead of individual cells. This results in a 36 × 1 feature vector for each block, and after computing the feature vectors for all blocks, they are combined to represent the entire image.

Using the feature vector generated by HOG and training it with SVM can result in a model that correctly distinguishes between humans or objects. The Dlib model is a landmark facial detector generated based on the above method, used to estimate the positions of 68 coordinates (x, y) that map facial points on a human face. These points are distributed as follows: chin (0–7), eyebrows (18–27), nose (28–36), eyes (37–48), and mouth (49–68). In order to improve accuracy in facial recognition, 13 additional points were added on the forehead (69–81) [26,27,28].

### 2.4. SpO2 Measurement Method

The calculation of blood oxygen saturation measurement [5] is shown in Equation (6):(6)$$SpO2=\frac{HbO_{2}}{HbO_{2}+Hb}\times 100\%$$*SpO2* stands for oxygen saturation, which refers to the strength of oxygen in the blood. *HbO2* is oxygenated hemoglobin, while *Hb* is deoxygenated hemoglobin. According to the Beer-Lambert law and the theory of light reflection, pulse oximeters commonly sold on the market use light waves with wavelengths of 660 nm and 940 nm to measure *SpO2*. The maximum variation in transmission intensity caused by the artery is determined by the most transmissive/reflected intensities of the two light waves ($I_{DC^{\lambda 1}}$ and $I_{DC^{\lambda 2}}$) and the maximum variable transmission intensity caused by the artery ($I_{AC^{\lambda 1}}$ and $I_{AC^{\lambda 2}}$) is shown in Equation (7) [5].
(7)$$SpO2=A\frac{I_{AC^{\lambda 1}}/I_{DC^{\lambda 1}}}{I_{AC^{\lambda 2}}/I_{DC^{\lambda 2}}}+B=A\times R+B$$The *A* and *B* are empirical coefficients determined by calibration accuracy, while *R* is the measured value. Generally, dual-wavelength pulse oximeters follow two principles, and the related values are shown in Figure 6 [5]:The difference in absorption coefficient between *HbO2* and *Hb* is larger at the same wavelength.The absorption coefficients of *HbO2* and *Hb* are similar.

As shown in the above figure, 440 nm, 520 nm, 805 nm, and 940 nm can all be used in combination with 660 nm for dual-wavelength calculations. Depending on the selected wavelengths, the correlation coefficients in the formula will also change.

## 3. Methodology

### 3.1. System Architecture 

Figure 7 shows the system architecture of this paper, and Figure 8 shows the system block diagram. In order to assist patients in monitoring their blood oxygen levels at any time without the need to purchase expensive oximeters, this paper employs a common web camera for measurement. A university staff member was used as the test subject. First, the test subject underwent blood oxygen measurement using a Taiwan FDA-certified oximeter. During this process, the system also performed face detection. After the test subject’s face was detected by the web camera, the red and blue light wave values on their forehead were captured within the region of interest. The required band was then extracted using a bandpass filter. The extracted light waves were used for calculations. To verify accuracy, the results were compared with the measurements from the oximeter.

### 3.2. Experimental Procedure

Figure 9 shows the flowchart of the non-contact blood oxygen measurement process in this paper. The non-contact blood oxygen measurement in this system can be divided into the following five steps:Use the faculty and students at a university as the subjects of this paper to measure the blood oxygen values of different age groups and genders.The tester uses a network camera to capture facial features and selects the forehead area of interest for subsequent calculations.Blood oxygen measurement requires comparison of two types of light waves. In this paper, red and blue light waves in the RGB three-color light wave are used for comparison.The extracted red and blue light waves are passed through a bandpass filter to eliminate waveform noise.The standard deviation and mean of the red and blue light waves after the bandpass filter are calculated, and the blood oxygen value is calculated through the blood oxygen formula. Finally, other factors that may cause experimental errors are discussed.

### 3.3. Red and Blue Light Wave Extraction

In this paper, blood oxygen calculation is performed on the forehead portion of the human face, and in order to calculate blood oxygen values, two color segments need to be extracted. This paper selected blue light with a similar absorption coefficient and red light with a large difference in absorption coefficient. To speed up the calculation, the red and blue light waves were extracted from RGB, as shown in Figure 10 and Figure 11, respectively. The X-axis in the figures represents the number of data, with the unit being “samples”, and the Y-axis representing the values of the red and blue light waves, with the unit being bits per pixel (bpp).

### 3.4. The Mean and Standard Deviation of Red and Blue Light Waves

The calculation of blood oxygen values in this paper is based on the red and blue light waves after bandpass filtering. A total of 100 data points were used to calculate the mean value, and the standard deviation was calculated using Equation (8):(8)$$\sigma =\sqrt{\frac{1}{N}\overset{N}{\underset{i=1}{\sum }}{\left(x_{i}-\bar{x}\right)}^{2}}$$

When the mean (*DC*) and standard deviation (*AC*) of the red-blue light waves are calculated, the standard deviation of the red and blue light waves needs to be divided by the mean. Then, the correlation coefficient *R* of blood oxygen is obtained by dividing the red light by the blue light, as shown in Equation (9).
(9)$$R=\frac{AC_{Red}/DC_{Red}}{AC_{Blue}/DC_{Blue}}$$

The blood oxygen formula used in this paper is shown in Equation (10):(10)$$SpO_{2}=A-B*R$$

In reference [5], red and green light waves were used for the paper. The empirical coefficients *A* and *B* were extracted by conducting multiple breath-holding contrast tests on multiple volunteer subjects under the same environmental temperature and lighting conditions. Thirty sets of data obtained from 30 different volunteers were used to calibrate the empirical coefficients *A* and *B*. In this paper, red and blue light waves were used for testing, and the blood oxygen calculation was obtained by adjusting the empirical coefficient *B* through testing on 10 subjects, as shown in Equation (11).
(11)$$SpO_{2}=125-28*R$$

## 4. Results 

### Experimental Results 

This paper measured the results of 30 subjects. While the testers conducted non-contact blood oxygen measurements, the blood oxygen machine was also used to measure the values, and the two measurement values were compared to investigate various factors that could cause errors in the experimental results. The tester needed to extract the RPPG signal reflected from the forehead’s light. In order to make the experimental results more accurate, the paper environment was set to a lighting intensity of 500 to 650 and a testing distance of 65 cm.

The performance indicator for this experiment is the mean absolute error (MAE), as shown in Equation (12), where$\;y_{i}$ is the actual output value of the *i*-th sample, $\hat{y_{i}}$ is the machine-measured output value of the *i*-th sample, and *n* is the total number of predicted samples.
(12)$$MAE=\frac{1}{n}\overset{n}{\underset{i=1}{\sum }}\left|y_{i}-{\hat{y}}_{l}\right|$$

Through the blood oxygen measurements of the same testers every three days, the data expansion effect was achieved, and the experimental values are shown in Table 2.

After measuring 30 subjects and expanding the data with an additional 30 experimental data points, the experimental values were plotted as a line graph against the machine-measured values of the blood oxygen machine, as shown in Table 2. The orange line represents the experiment data, and the blue line represents the device data. The X-axis represents the number of testers, and the Y-axis represents the values of blood oxygen saturation, as shown in Figure 12. From these 60 test data points, the mean absolute error (MAE) of this paper was found to be 0.62%.

## 5. Discussion

This paper extracted the RPPG signal reflected from the forehead’s light, and therefore, there are many external factors that can affect the research values. These factors were investigated and discussed.

### 5.1. Light Intensity 

As this paper’s experimental method is based on the principle of light reflection, the strength of the light indirectly affects the experiment’s values. Therefore, three different light intensity environments were tested to investigate their effects. With the experimental distance fixed at 60 cm, the first environment tested had a normal light intensity of 550 lux, followed by 220 lux and gradually dimming the light until it reached 100 lux. It was found that the accuracy was higher under the 550 lux light intensity.

### 5.2. Experimental Distance 

In terms of the experimental distance, if the forehead area for signal extraction is too far from the camera, the captured area will be smaller, and this will cause errors in the reflected light values. In this paper, the light intensity was fixed at 550 lux, and the experimental results were tested at distances of 30 cm, 60 cm, and 90 cm. The experimental results showed that the detection accuracy was higher at a distance of 60 cm.

### 5.3. Skin Color 

This paper investigated the impact of skin color on the RPPG signal. The experiments were conducted under a fixed illumination of 550 lux and a testing distance of 60 cm. The results showed that there was no difference in accuracy between individuals with yellow and black skin, at a similar age.

### 5.4. Comparison with Related Literature

In [16], the same red-blue channel was used for calculating blood oxygen saturation as in this paper, but the formula for blood oxygen calculation was different, as shown in Equation (13), where the blood oxygen-related measurement value R is the same as in Equation (7).
(13)SpO2=97.61+0.42 x R

Due to the coefficient problem in the formula, the calculation result of blood oxygen saturation will always be higher than 97. If a patient with a blood oxygen saturation below 97 appears, it will cause errors in the experiment. The average error of the experiment is 1.4. Therefore, compared with previous studies, this paper is more accurate. Table 3 is the comparison table between this paper and reference [29].

In Reference [30], red and green light waves were used for dual-wavelength calculation of blood oxygen saturation. The extracted red and green light waves were processed through independent component analysis (ICA) to capture clear low-frequency and high-frequency ranges. According to their experimental results, the average error was 3.02%. In terms of image processing, under similar hardware devices, Reference [30] took a total of 105 s for blood oxygen saturation calculation, while in this paper, it only took 30 s. Table 4 is presented below for comparison.

## 6. Conclusions

This paper implemented a low-cost system for non-contact blood oxygen saturation measurement using a commercially available webcam to detect facial features and measure physiological parameters. This system assists users in measuring their blood oxygen saturation in real-time, saving the cost of purchasing a blood oxygen testing machine and providing convenience to the user.

To prove the accuracy of this system, the research methodology and system design were tested at distances of 30 cm, 50 cm, 125 cm, and 190 cm, and under different lighting conditions such as light levels below 250 lumens, between 250 and 650 lumens, and above 650 lumens. To supplement the limited experimental data, this paper used the daily blood oxygen values of six test subjects for data expansion. The results of this experiment confirmed that the values in the region of interest on the forehead, after signal processing and blood oxygen calculation formula, had an average absolute error of 0.62%.

Compared to Bhattacharjee’s research, which cannot measure blood oxygen values below 97%, and has an experimental error of 1.4%, this paper’s blood oxygen calculation formula and error are more accurate. Compared to Al-Naji’s research, which has an average absolute error of 3.02%, this paper took only 30 s to measure blood oxygen saturation values, which is faster and has better accuracy.

With the popularization of smart devices, the algorithm can be written as an app or API and installed on devices with cameras, such as smartphones, tablets, or laptops, allowing people to monitor their physical condition at any time. However, since it has not yet been approved by the FDA for use in real clinical situations, it cannot be applied in medical institutions. Nonetheless, it can be applied in home care, community care, remote health care and other fields, and can also improve the quality of care for COVID-19 patients and reduce the burden on health care.

## Figures and Tables

**Figure 1 bioengineering-10-00524-f001:**
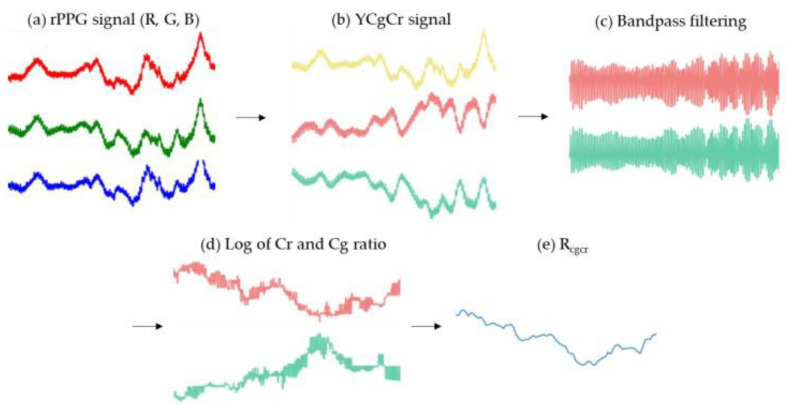
YCgCr waveform display. Process of extracting *R_cgcr_* value from rPPG signal (**a**) R, G, B signals extracted from face ROI. The red line is the R raw signal, the green line is the G raw signal, and the blue line is B raw signal. (**b**) R, G, B signals are converted into YCgCr color space. The yellow line is the Y signal, the pink line is the Cr signal, and the light blue line is the Cg signal. (**c**) Result of bandpass filtering on Cg and Cr signals. The pink line is the filtered Cr signal, the light blue line is the filtered Cg signal. (**d**) The result of is, the log value of the peak/valley extracted from the Cg and Cr signals to which bandpass filtering is applied, respectively. The pink line is the calculated Cr signal, the light blue line is the calculated Cg signal. (**e**) The result is, Cr ratio/Cg ratio.

**Figure 2 bioengineering-10-00524-f002:**
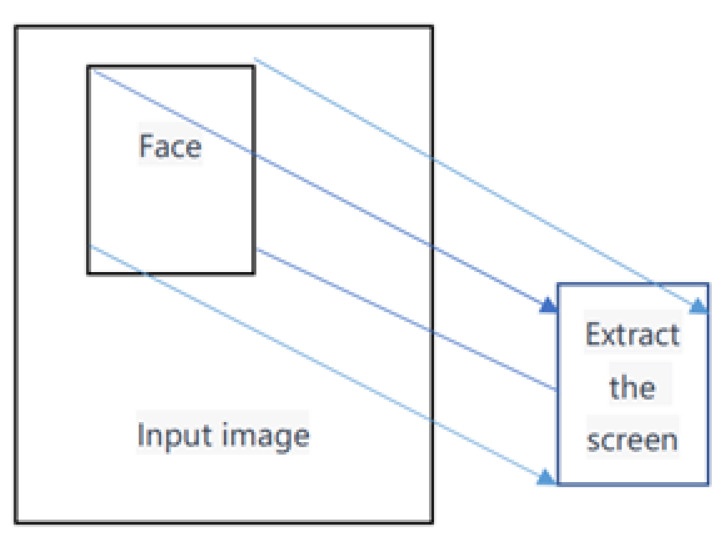
ROI image extraction.

**Figure 3 bioengineering-10-00524-f003:**
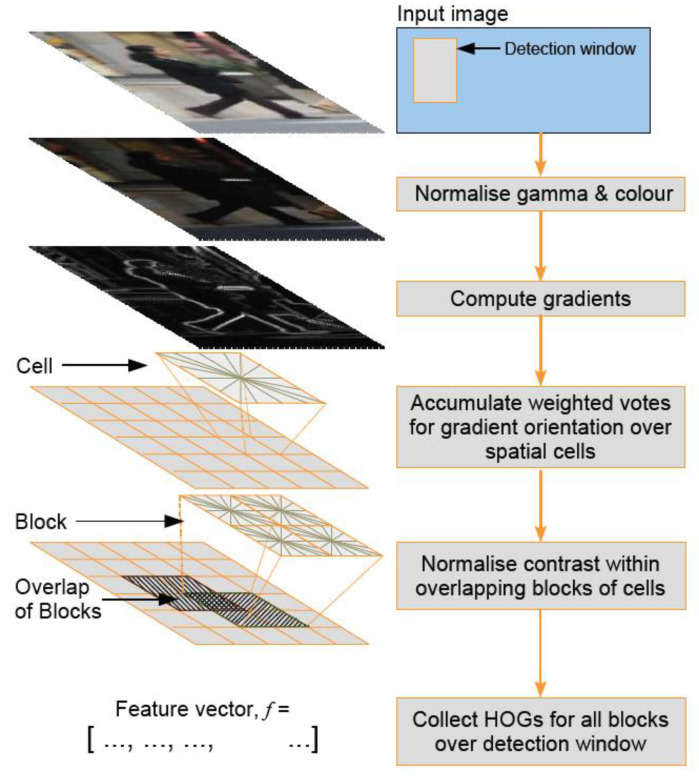
HOG calculation flowchart [20].

**Figure 4 bioengineering-10-00524-f004:**
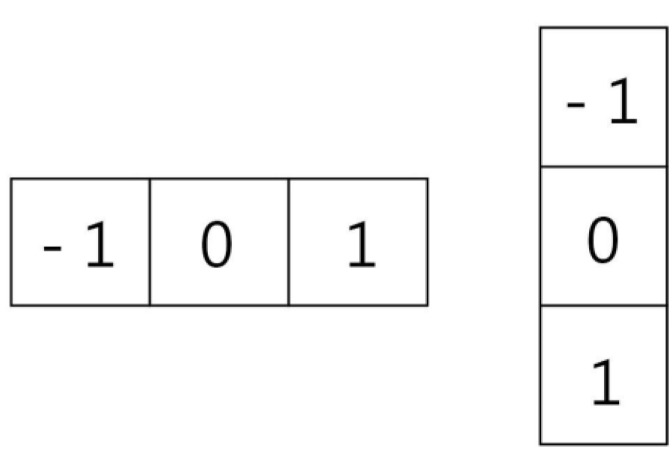
Gradient calculation [23].

**Figure 5 bioengineering-10-00524-f005:**
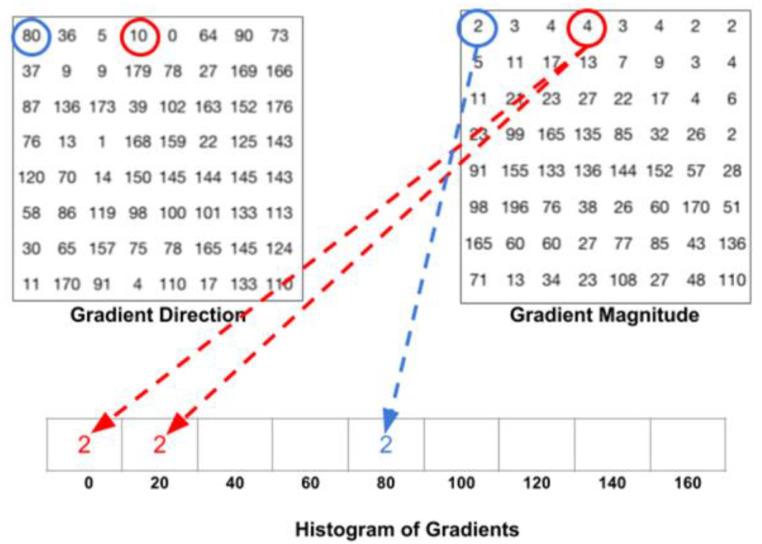
Gradient histogram.

**Figure 6 bioengineering-10-00524-f006:**
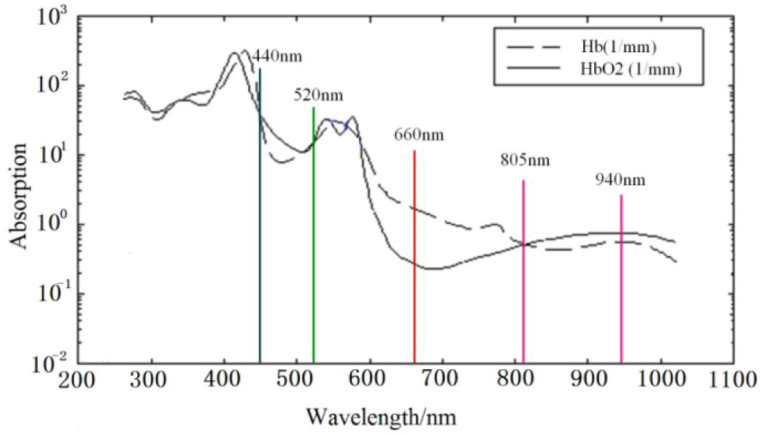
The absorption spectrum of HbO2 and Hb.

**Figure 7 bioengineering-10-00524-f007:**
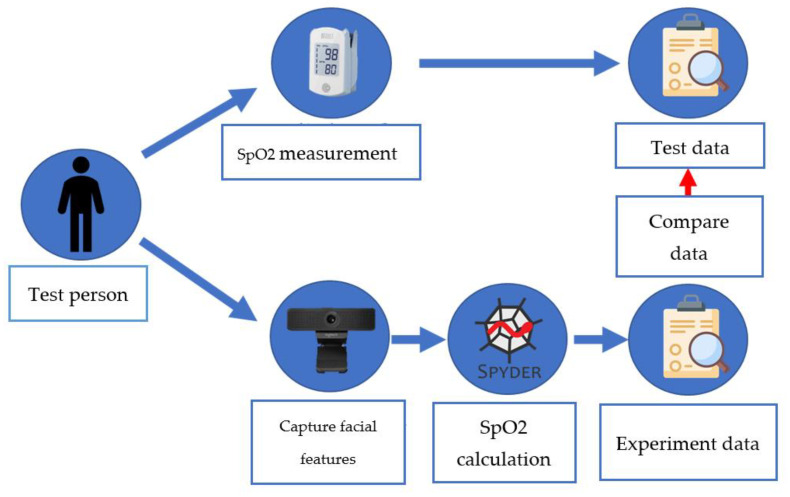
Architecture diagram of SpO2 measurement system.

**Figure 8 bioengineering-10-00524-f008:**
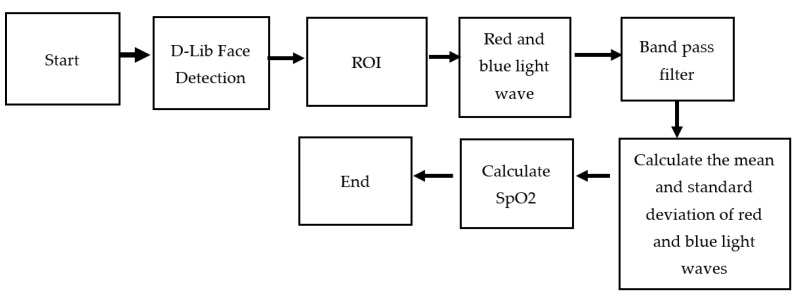
Block diagram of SpO2 measurement system.

**Figure 9 bioengineering-10-00524-f009:**
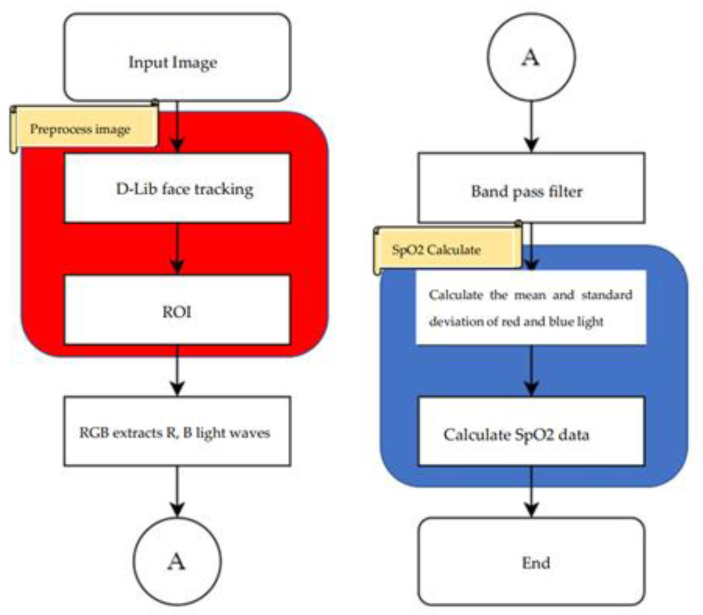
Non-contact SpO2 measurement flow chart.

**Figure 10 bioengineering-10-00524-f010:**
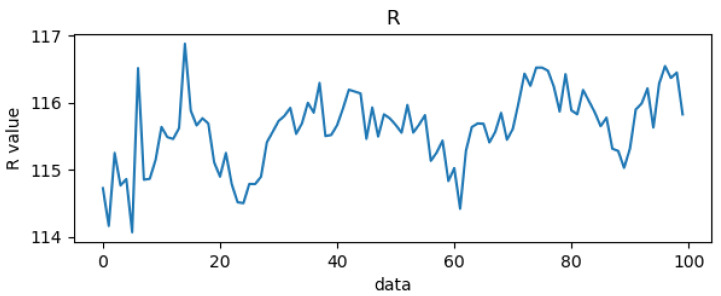
Red light wave.

**Figure 11 bioengineering-10-00524-f011:**
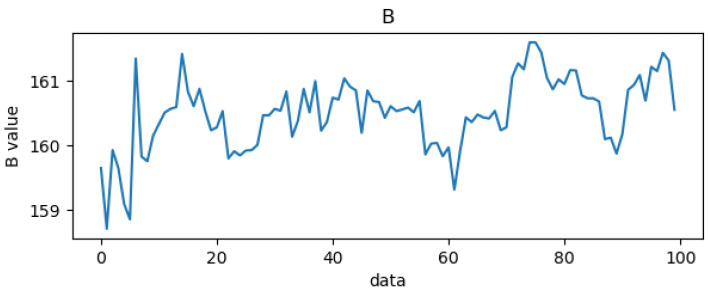
Blue light wave.

**Figure 12 bioengineering-10-00524-f012:**
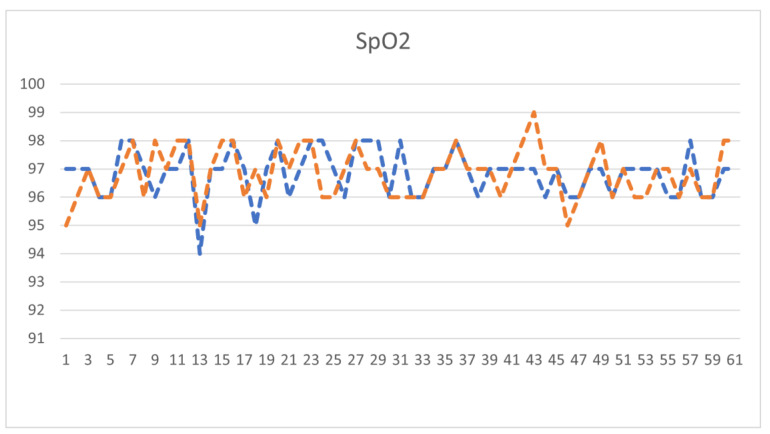
Blood oxygen data line chart.

**Table 1 bioengineering-10-00524-t001:** Standard for blood oxygen saturation.

Blood Oxygen Saturation (SpO2)	Executive Actions
≥95%	Normal blood oxygen saturation
91–94%	Please be aware of the blood oxygen levels in this range and consider going to the hospital for a check-up.
≤90%	If blood oxygen saturation is too low, immediate emergency treatment or hospitalization is necessary.

Source: Health unit of the government of New York State in the United States [15].

**Table 2 bioengineering-10-00524-t002:** Daily observations.

	Tester	1	2	3	4	5	6
Day 1	Experimental data	96	96	97	97	98	97
	SpO2 device data	96	96	97	97	98	97
Day 2	Experimental data	96	97	97	97	97	97
	SpO2 device data	97	97	96	97	98	99
Day 3	Experimental data	96	97	96	96	97	97
	SpO2 device data	97	97	95	96	97	98
Day 4	Experimental data	96	97	97	97	97	96
	SpO2 device data	96	97	96	96	97	97
Day 5	Experimental data	96	98	96	96	97	97
	SpO2 device data	96	97	96	96	98	98

**Table 3 bioengineering-10-00524-t003:** SpO2 experiment comparison table.

	SpO2 Device Data	Bhattacharjee [29]	This Research
Experimental data	99	98	97
	97	98	97
	95	99	94
	98	98	98
	97	98	98
	96	97	96
Mean absolute error		1.4%	0.62%

**Table 4 bioengineering-10-00524-t004:** Time comparison table for SpO2 detection.

	Al-Naji [30]	This Research
Light wave use	Red, Green	Red, Blue
Language use	Matlab	Python
Average error	3.02%	0.62%
Measure time	105 Sec.	30 Sec.

## Data Availability

Not applicable.

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
