# Peer review of "Using Contactless Facial Image Recognition Technology to Detect Blood Oxygen Saturation"

_bioengineering, 2023, doi:10.3390/bioengineering10050524_

Round 1

Reviewer 1 Report

Very intersting paper showing the interest and efficacy of a non contact SpO2 measurement device. The paper clearly demonstrate the interest. Nevertheless we have any idea of the efficacy in real clinical condition and the performance of the device in ICU context with intubation, profound destination shock and sepsis.

I should be discuss in the text regarding potential applications.

Reviewer 2 Report

Since the outbreak of COVID-19, as of January 2023, there have been over 670 million cases and more than 6.8 million deaths worldwide. Infections can cause inflammation in the lungs and decrease blood oxygen levels, which can lead to breathing difficulties and endanger life. As the situation continues to escalate, non-contact machines are used to assist patients at home to monitor their blood oxygen levels without encountering others.

The authors propose a system based on a general network camera to capture the forehead area through facial recognition software, and then processes the red and blue light waveforms to calculate blood oxygen saturation using the principle of light wave reflection and investigates the effect of illumination on experimental values. The experimental results of this paper were compared with a blood oxygen meter certified by the Ministry of Health and Welfare in Taiwan, and the experimental results had only a maximum error of 2%, which is better than the 3% to 5% error rates in other literature. In terms of measurement time, when using similar equipment to other literature, the measurement time was only 30 seconds, which is better than the one minute in other literature.

Authors concluded that their paper not only saves a lot of equipment expenses but also provides convenience and safety for those who need to monitor their blood oxygen levels at home.

Interesting study.

I have only minor suggestions for the authors:

1.     Revise the abstract better summarizing the sections

2.     Insert a clear purpose. Now it is lacking

3.     Figure must be described in details

4.     I suggest to separate the discussion and to insert here also the limitations

5.     Usually the conclusions do not contain citations

Reviewer 3 Report

This is an interesting and informative paper on using a general network camera to monitor blood oxygen levels at home during the COVID-19 pandemic. Here are a few comments and suggestions for improvement:

Clarify the purpose and significance of the study: In the introduction, it would be helpful to explain why monitoring blood oxygen levels is important for COVID-19 patients and why non-contact machines are necessary for home monitoring. You could also discuss the limitations of existing equipment and methods for measuring blood oxygen saturation.

Discuss the implications of the results: The conclusion briefly mentions the benefits of the proposed method, but it would be helpful to discuss the implications of the results in more detail. For example, you could discuss how this method could be used to improve the quality of care for COVID-19 patients and how it could potentially reduce the burden on healthcare systems.

Provide more information on the experimental values: In the abstract, you mention that the experimental results had a maximum error of 2%, which is better than other literature. However, it would be helpful to provide more information on the range of error values across the measurements and how these compare to the error values reported in other literature.

Overall, this is a promising study that could have important implications for home monitoring of COVID-19 patients. By addressing the above suggestions, you could make the paper even more informative and impactful.
